# Sliced-Wasserstein Auto-Encoders

**Soheil Kolouri, Phillip E. Pope, & Charles E. Martin,**
Information and Systems Sciences Laboratory
HRL Laboratories, LLC.
Malibu, CA, USA
{skolouri,pepope,cemartin}@hrl.com

**Gustavo K. Rohde**
Department of Electrical Engineering
University of Virginia
Charlottesville, VA, USA
gustavo@virginia.edu

## Abstract

In this paper we use the geometric properties of the optimal transport (OT) problem and the Wasserstein distances to define a prior distribution for the latent space of an auto-encoder. We introduce Sliced-Wasserstein Auto-Encoders (SWAE), that enable one to shape the distribution of the latent space into any samplable probability distribution without the need for training an adversarial network or having a likelihood function specified. In short, we regularize the auto-encoder loss with the sliced-Wasserstein distance between the distribution of the encoded training samples and a samplable prior distribution. We show that the proposed formulation has an efficient numerical solution that provides similar capabilities to Wasserstein Auto-Encoders (WAE) and Variational Auto-Encoders (VAE), while benefiting from an embarrassingly simple implementation. We provide extensive error analysis for our algorithm, and show its merits on three benchmark datasets.

Scalable generative models that capture the rich and often nonlinear distribution of high-dimensional data, (i.e., image, video, and audio), play a central role in various applications of machine learning, including transfer learning Isola et al. (2017); Murez et al. (2018), super-resolution Ledig et al. (2016); Kolouri & Rohde (2015), image inpainting and completion Yeh et al. (2017), and image retrieval Creswell & Bharath (2016), among many others. The recent parametric generative models, including Generative Adversarial Networks (GANs) Goodfellow et al. (2014); Radford et al. (2015); Arjovsky et al. (2017); Berthelot et al. (2017) and Variational auto-encoders (VAE) Kingma & Welling (2013); Mescheder et al. (2017); Bousquet et al. (2017) enable an unsupervised and end-to-end modeling of the high-dimensional distribution of the training data.

Learning such generative models boils down to minimizing a dissimilarity measure between the data distribution and the output distribution of the generative model. To this end, and following the work of Arjovsky et al. (2017) and Bousquet et al. (2017), we approach the problem of generative modeling from the optimal transport point of view. The optimal transport problem Villani (2008); Kolouri et al. (2017) provides a way to measure the distances between probability distributions by transporting (i.e., morphing) one distribution into another. Moreover, and as opposed to the common information theoretic dissimilarity measures (e.g., $f$-divergences), the p-Wasserstein dissimilarity measures that arise from the optimal transport problem: 1) are true distances, and 2) metrize a weak convergence of probability measures (at least on compact spaces). Wasserstein distances have recently attracted a lot of interest in the learning community Frogner et al. (2015); Gulrajani et al. (2017); Bousquet et al. (2017); Arjovsky et al. (2017); Kolouri et al. (2017) due to their exquisite geometric characteristics Santambrogio (2015). See the supplementary material for an intuitive example showing the benefit of the Wasserstein distance over commonly used $f$-divergences.

In this paper, we introduce a new type of auto-encoders for generative modeling (Algorithm 1), which we call Sliced-Wasserstein auto-encoders (SWAE), that minimize the sliced-Wasserstein distance between the distribution of the encoded samples and a samplable prior distribution. Our work is most closely related to the recent work by Bousquet et al. (2017) and more specifically the follow-up work by Tolstikhin et al. (2017). However, our approach avoids the need to perform adversarial training in the encoding space and is not restricted to closed-form distributions, while still benefiting from a Wasserstein-like distance measure in the latent space. Calculating the Wasserstein distance can be computationally expensive, but our approach permits a simple numerical solution to the problem. Finally, we note that there has been several concurrent papers, including the work by Deshpande et al. (2018) and Şimşekli et al. (2018), that also looked into the application of sliced-Wasserstein distance

in generative modeling. Regardless of the concurrent nature of these papers, our work remains novel and is distinguished from these methods. Deshpande et al. (2018) use the sliced-Wasserstein distance to match the distributions of high-dimensional reconstructed images, which require large number of slices, $\mathcal{O}(10^4)$, while in our method and due to the distribution matching in the latent space we only need $\mathcal{O}(10)$ slices. We also note that Deshpande et al. (2018) proposed to learn discriminative slices to mitigate the need for a very large number of random projections that is in essence similar to the adversarial training used in GANs, which contradicts with our goal of not using adversarial training. Şimşekli et al. (2018), on the other hand, take an interesting but different approach of parameter-free generative modeling via sliced-Wasserstein flows.

## 1 NOTATION AND PRELIMINARIES

Let $X$ denote the compact domain of a manifold in Euclidean space and let $x_n \in X$ denote an individual input data point. Furthermore, let $\rho_X$ be a Borel probability measure defined on $X$. We define the probability density function $p_X(x)$ for input data $x$ to be:

$$d\rho_X(x) = p_X(x)dx$$

Let $\phi : X \to Z$ denote a deterministic parametric mapping from the input space to a latent space $Z$ (e.g., a neural network encoder). To obtain the density of the push forward of $\rho_X$ with respect to $\phi$, i.e., $\rho_Z = \phi_*(\rho_X)$, we use Random Variable Transformation (RVT) Gillespie (1983)). In short, the probability density function of the encoded samples $z$ can be expressed in terms of $\phi$ and $p_X$ by:

$$p_Z(z) = \int_X p_X(x)\delta(z - \phi(x))dx, \tag{1}$$

where $\delta$ denotes the Dirac distribution function. Similar to variational Auto-Encoders (VAEs) Kingma & Welling (2013) and the Wasserstein Auto-Encoders (WAE) Tolstikhin et al. (2017), our main objective is to encode the input data points $x \in X$ into latent codes $z \in Z$ such that: 1) $x$ can be recovered/approximated from $z$, and 2) the probability density function of the encoded samples, $p_Z$, follows a prior distribution $q_Z$. Let $\psi : Z \to X$ be the decoder that maps the latent codes back to the original space such that

$$p_Y(y) = \int_X p_X(x)\delta(y - \psi(\phi(x)))dx, \tag{2}$$

where $y$ denotes the decoded samples. It is straightforward to see that when $\psi = \phi^{-1}$ (i.e. $\psi(\phi(\cdot)) = id(\cdot)$), the distribution of the decoder $p_Y$ and the input distribution $p_X$ are identical. Hence, in its most general form, the objective of such auto-encoders simplifies to learning $\phi$ and $\psi$, so that they minimize a dissimilarity measure between $p_Y$ and $p_X$, and between $p_Z$ and $q_Z$. In what follows, we briefly review the existing dissimilarity measures for these distributions.

### 1.1 MINIMIZING DISSIMILARITY BETWEEN $p_X$ AND $p_Y$

We first emphasize that the VAE often assumes stochastic encoders and decoders Kingma & Welling (2013), while we consider the case of only deterministic mappings. Although, we note that, similar to WAE, SWAE can also be formulated with stochastic encoders. Different measures have been used previously to compute the dissimilarity between $p_X$ and $p_Y$. Most notably, Nowozin et al. (2016) showed that for the general family of $f$-divergences, $D_f(p_X, p_Y)$, (including the KL-divergence, Jensen-Shannon, etc.), using the Fenchel conjugate of the convex function $f$ and minimizing $D_f(p_X, p_Y)$ leads to a min-max problem that is equivalent to the *adversarial training* widely used in the generative modeling literature Goodfellow et al. (2014); Makhzani et al. (2015); Mescheder et al. (2017).

Others have utilized the rich mathematical foundation of the OT problem and Wasserstein distances Arjovsky et al. (2017); Gulrajani et al. (2017); Bousquet et al. (2017); Tolstikhin et al. (2017) to define a distance between $p_X$ and $p_Y$. In Wasserstein-GAN, Arjovsky et al. (2017) utilized the Kantorovich-Rubinstein duality for the 1-Wasserstein distance, $W_1(p_X, p_Y)$, and reformulated the problem as a min-max optimization that is solved through an adversarial training scheme.

Inspired by the work of Bousquet et al. (2017) and Tolstikhin et al. (2017), it can be shown that (see supplementary material for a proof):

$$W_c(p_X, p_Y) \leq W_c^{\ddagger}(p_X, p_Y) \quad := \quad \mathbb{E}_{p_X}(c(x, \psi(\phi(x)))) \tag{3}$$
$$= \quad \int_X c(x, \psi(\phi(x)))p_X(x)dx,$$

Furthermore, the r.h.s. of equation 3 supports a simple implementation where for i.i.d samples of the input distribution, $\{x_n\}_{n=1}^{N}$, the upper bound can be approximated as:

$$W_c^{\ddagger}(p_X, p_Y) \approx \frac{1}{N} \sum_{n=1}^{N} c(x_n, \psi(\phi(x_n))) \tag{4}$$

The r.h.s of equation 3 and equation 4 take advantage of the existence of pairs $x_n$ and $y_n = \psi(\phi(x_n))$, which make $f(\cdot) = \psi(\phi(\cdot))$ a transport map between $p_X$ and $p_Y$ (but not necessarily the optimal transport map). In this paper, we minimize $W_c^{\ddagger}(p_X, p_Y)$ following equation 4 to minimize the discrepancy between $p_X$ and $p_Y$. Next, we focus on the discrepancy measures between $p_Z$ and $q_Z$.

## 1.2 MINIMIZING DISSIMILARITY BETWEEN $p_Z$ AND $q_Z$

If $q_Z$ is a known distribution with an explicit formulation (e.g. Normal distribution) the most straightforward approach for measuring the (dis)similarity between $p_Z$ and $q_Z$ is the log-likelihood of $z = \phi(x)$ with respect to $q_Z$, formally:

$$sup_\phi \int_X p_X(x) log(q_Z(\phi(x))) dx \tag{5}$$

maximizing the log-likelihood is equivalent to minimizing the KL-divergence between $p_Z$ and $q_Z$, $D_{KL}(p_Z, q_Z)$ (see supplementary material for more details and derivation of Equation equation 5). This approach has two major limitations: 1) The KL-Divergence and in general $f$-divergences do not provide meaningful dissimilarity measures for distributions supported on non-overlapping low-dimensional manifolds Arjovsky et al. (2017); Kolouri et al. (2018) (see supplementary material), which is common in hidden layers of neural networks, and therefore they do not provide informative gradients for training $\phi$, and 2) we are limited to distributions $q_Z$ that have known explicit formulations, which is restrictive as it eliminates the ability to use the much broader class of samplable distributions.

Various alternatives exist in the literature to address the above-mentioned limitations. These methods often sample $\tilde{\mathcal{Z}} = \{\tilde{z}_j\}_{j=1}^{N}$ from $q_Z$ and $\mathcal{Z} = \{z_n = \phi(x_n)\}_{n=1}^{N}$ from $p_X$ and measure the discrepancy between these sets (i.e. point clouds). Note that there are no one-to-one correspondences between $\tilde{z}_j$s and $z_n$s. In their influential WAE paper, Tolstikhin et al. (2017) proposed two different approaches for measuring the discrepancy between $\tilde{\mathcal{Z}}$ and $\mathcal{Z}$, namely the GAN-based and the *maximum mean discrepancy* (MMD)-based approaches. The GAN-based approach proposed in Tolstikhin et al. (2017) defines a discriminator network, $D_Z(p_Z, q_Z)$, to classify $\tilde{z}_j$s and $z_n$s as coming from 'true' and 'fake' distributions correspondingly, and proposes a min-max adversarial optimization for learning $\phi$ and $D_Z$. The MMD-based approach, utilizes a positive-definite reproducing kernel $k : Z \times Z \to \mathbb{R}$ to measure the discrepancy between $\tilde{\mathcal{Z}}$ and $\mathcal{Z}$. The choice of the kernel and its parameterization, however, remain a data-dependent design parameter.

An interesting alternative approach is to use the Wasserstein distance between $p_Z$ and $q_Z$. Following the work of Arjovsky et al. (2017), this can be accomplished utilizing the Kantorovich-Rubinstein duality and through introducing a min-max problem, which leads to yet another adversarial training scheme similar to the GAN-based method in Tolstikhin et al. (2017). Note that, since elements of $\tilde{\mathcal{Z}}$ and $\mathcal{Z}$ are not paired, an approach similar to equation 4 could not be used to minimize the discrepancy. In this paper, we propose to use the sliced-Wasserstein metric, Rabin & Peyré (2011); Rabin et al. (2011); Bonneel et al. (2015); Kolouri et al. (2016b); Carriere et al. (2017); Kolouri et al. (2018), to measure the discrepancy between $p_Z$ and $q_Z$. We show that using the sliced-Wasserstein distance ameliorates the need for training an adversary network or choosing a data-dependent kernel (as in WAE-MMD), and provides an efficient, stable, and simple numerical implementation.

Before explaining our proposed approach, it is worthwhile to point out the major difference between learning auto-encoders as generative models and GANs. In GANs, one needs to minimize a distance between $\{\psi(\tilde{z}_j)|\tilde{z}_j \sim q_Z\}_{j=1}^{M}$ and $\{x_n\}_{n=1}^{M}$, which are high-dimensional point clouds for which there are no correspondences between $\psi(\tilde{z}_j)$s and $x_n$s. For the auto-encoders, on the other hand, there exists correspondences between the high-dimensional point clouds $\{x_n\}_{n=1}^{M}$ and $\{y_n = \psi(\phi(x_n))\}_{n=1}^{M}$, and the problem simplifies to matching the lower-dimensional point clouds $\{\phi(x_n)\}_{n=1}^{M}$ and $\{\tilde{z}_j \sim q_Z\}_{j=1}^{M}$. In other words, the encoder performs a nonlinear dimensionality reduction, that enables us to solve a simpler problem compared to GANs. Next we introduce the details of our approach.

## 2 PROPOSED METHOD

In what follows we first provide a brief review of the necessary equations to understand the Wasserstein and sliced-Wasserstein distances and then present our Sliced Wasserstein auto-encoder (SWAE).

### 2.1 WASSERSTEIN DISTANCES

The Wasserstein distance between probability measures $\rho_X$ and $\rho_Y$, with corresponding densities $d\rho_X = p_X(x)dx$ and $d\rho_Y = p_Y(y)dy$ is defined as:

$$W_c(p_X, p_Y) = inf_{\gamma \in \Gamma(\rho_X, \rho_Y)} \int_{X \times Y} c(x, y)d\gamma(x, y) \qquad (6)$$

where $\Gamma(\rho_X, \rho_Y)$ is the set of all transportation plans (i.e. joint measures) with marginal densities $p_X$ and $p_Y$, and $c : X \times Y \to \mathbb{R}^+$ is the transportation cost. equation 6 is known as the Kantorovich formulation of the optimal mass transportation problem, which seeks the optimal transportation plan between $p_X$ and $p_Y$. If there exist diffeomorphic mappings, $f : X \to Y$ (i.e. transport maps) such that $y = f(x)$ and consequently,

$$p_Y(y) = \int_X p_X(x)\delta(y - f(x))dx \xrightarrow[\text{a diffeomorphism}]{\text{When } f \text{ is}} p_Y(y) = det(Df^{-1}(y))p_X(f^{-1}(y)) \qquad (7)$$

where $det(D\cdot)$ is the determinant of the Jacobian, then the Wasserstein distance could be defined based on the Monge formulation of the problem (see Villani (2008) and Kolouri et al. (2017)) as:

$$W_c(p_X, p_Y) = min_{f \in MP} \int_X c(x, f(x))d\rho_X(x) \qquad (8)$$

where $MP$ is the set of all diffeomorphisms that satisfy equation 7. As can be seen from equation 6 and equation 8, obtaining the Wasserstein distance requires solving an optimization problem. We note that various efficient optimization techniques have been proposed in the past (e.g. Cuturi (2013); Solomon et al. (2015); Oberman & Ruan (2015)) to solve this optimization. For one-dimensional probability densities, $p_X$ and $p_Y$, however, the Wasserstein distance has a closed-form solution. Let $P_X$ and $P_Y$ be the cumulative distributions of one-dimensional probability distributions $p_X$ and $p_Y$, correspondingly. The Wassertein distance can then be calculated as below (see Kolouri et al. (2017) for more details):

$$W_c(p_X, p_Y) = \int_0^1 c(P_X^{-1}(\tau), P_Y^{-1}(\tau))d\tau, \qquad (9)$$

This closed-form solution motivates the definition of sliced-Wasserstein distances.

### 2.2 SLICED-WASSERSTEIN DISTANCES

Sliced-Wasserstein distance has similar qualitative properties to the Wasserstein distance, but it is much easier to compute. The sliced-Wasserstein distance was used in Rabin & Peyré (2011); Rabin et al. (2011) to calculate barycenter of distributions and point clouds. Bonneel et al. (2015) provided a nice theoretical overview of barycenteric calculations using the sliced-Wasserstein distance. Kolouri et al. (2016b) used it to define positive definite kernels for distributions and Carriere et al. (2017) to define a kernel for persistence diagrams. Sliced-Wasserstein was recently used for learning Gaussian mixture models in Kolouri et al. (2018), and it was also used as a measure of goodness of fit for GANs in Karras et al. (2017).

The main idea behind the sliced-Wasserstein distance is to slice (i.e., project) higher-dimensional probability densities into sets of one-dimensional marginal distributions and compare these marginal distributions via the Wasserstein distance. The slicing/projection process is related to the field of Integral Geometry and specifically the Radon transform (see Helgason (2011)). The relevant result to our discussion is that a d-dimensional probability density $p_X$ can be uniquely represented as the set of its one-dimensional marginal distributions following the Radon transform and the Fourier slice theorem Helgason (2011). These one dimensional marginal distributions of $p_X$ are defined as:

$$\mathcal{R}p_X(t; \theta) = \int_X p_X(x)\delta(t - \theta \cdot x)dx, \ \ \forall \theta \in \mathbb{S}^{d-1}, \ \forall t \in \mathbb{R} \qquad (10)$$

where $\mathbb{S}^{d-1}$ is the d-dimensional unit sphere. Note that for any fixed $\theta \in \mathbb{S}^{d-1}$, $\mathcal{R}p_X(\cdot; \theta)$ is a one-dimensional slice of distribution $p_X$. In other words, $\mathcal{R}p_X(\cdot; \theta)$ is a marginal distribution of $p_X$

that is obtained from integrating $p_X$ over the hyperplane orthogonal to $\theta$. Utilizing these marginal distributions in equation 10, the sliced Wasserstein distance could be defined as:

$$SW_c(p_X, p_Y) = \int_{\mathbb{S}^{d-1}} W_c(\mathcal{R}p_X(\cdot; \theta), \mathcal{R}p_Y(\cdot; \theta)) d\theta \tag{11}$$

Given that $\mathcal{R}p_X(\cdot; \theta)$ and $\mathcal{R}p_Y(\cdot; \theta)$ are one-dimensional, the Wasserstein distance in the integrand has a closed-form solution (see equation 9). Moreover, it can be shown that $SW_c$ is a true metric (Bonnotte (2013) and Kolouri et al. (2016a)), and it induces the same topology as $W_c$, at least on compact sets Santambrogio (2015). A natural transportation cost that has extensively studied in the past is the $\ell_2^2$, $c(x, y) = \|x - y\|_2^2$, for which there are theoretical guarantees on existence and uniqueness of transportation plans and maps (see Santambrogio (2015) and Villani (2008)). When $c(x, y) = \|x - y\|_p^p$ for $p \geq 2$, the following upper bound hold for the SW distance:

$$SW_p^p(p_X, p_Y) \leq \alpha_{d,p} W_p^p(p_X, p_Y) \tag{12}$$

where, $\alpha_{d,p} = \frac{1}{d} \int_{\mathbb{S}^{d-1}} \|\theta\|_p^p d\theta \leq 1$. Chapter 5 in Bonnotte (2013) proves this inequality. In our paper, we are interested in $p = 2$, for which $\alpha_{p,d} = \frac{1}{d}$, and we have:

$$SW_2(p_X, p_Y) \leq \frac{1}{\sqrt{d}} W_2(p_X, p_Y) \tag{13}$$

In the Numerical Implementation Section, we provide a numerical experiment to compare $W_2$ and $SW_2$, that confirms the above equation.

### 2.3 SLICED-WASSERSTEIN AUTO-ENCODER (SWAE)

Our proposed formulation for the SWAE is as follows:

$$\operatorname{argmin}_{\phi, \psi} W_c^\ddagger(p_X, p_Y) + \lambda SW_c(p_Z, q_Z) \tag{14}$$

where $\phi$ is the encoder, $\psi$ is the decoder, $p_X$ is the data distribution, $p_Y$ is the data distribution after encoding and decoding ( equation 2), $p_Z$ is the distribution of the encoded data ( equation 1), $q_Z$ is a predefined samplable distribution, and $\lambda$ indicates the relative importance of the loss functions. To further clarify why we use the *sliced-Wasserstein* distance to measure the difference between $p_Z$ and $q_Z$, we reiterate that due to the lack of correspondences between $\tilde{z}_i$s and $z_j$s, one cannot minimize the upper-bound in equation 4, and calculation of the Wasserstein distance requires an additional optimization step to obtain the optimal coupling between $p_Z$ and $q_Z$. To avoid this additional optimization, while maintaining the favorable characteristics of the Wasserstein distance, we use the sliced-Wasserstein distance to measure the discrepancy between $p_Z$ and $q_Z$.

## 3 NUMERICAL IMPLEMENTATION

We now describe the numerical details of our approach.

### 3.1 NUMERICAL IMPLEMENTATION OF THE WASSERSTEIN DISTANCE IN 1D

The Wasserstein distance between two one-dimensional probability densities $p_X$ and $p_Y$ is obtained from equation 9. The integral in equation 9 can be numerically estimated using the midpoint Riemann sum, $\frac{1}{M} \sum_{m=1}^{M} a_m$, where $a_m = c(P_X^{-1}(\tau_m), P_Y^{-1}(\tau_m))$ and $\tau_m = \frac{2m-1}{2M}$ (see Fig. 1). In scenarios where only samples from the distributions are available, $x_m \sim p_X$ and $y_m \sim p_Y$, the empirical densities can be estimated as $p_X \approx p_{X,M} = \frac{1}{M} \sum_{m=1}^{M} \delta_{x_m}$ and $p_Y \approx p_{Y,M} = \frac{1}{M} \sum_{m=1}^{M} \delta_{y_m}$, where $\delta_{x_m}$ is the Dirac delta function centered at $x_m$. Therefore the corresponding empirical distribution function of $p_X$ is $P_X(t) \approx P_{X,M}(t) = \frac{1}{M} \sum_{m=1}^{M} u(t - x_m)$ where $u(.)$ is the step function ($P_{Y,M}(t)$ is defined similarly). From Glivenko-Cantelli Theorem we have that $\sup_t |P_{X,M}(t) - P_X(t)| \xrightarrow{a.s.} 0$, where the convergence behavior is achieved via Dvoretzky–Kiefer–Wolfowitz inequality bound: $Prob(\sup_t |P_{X,M}(t) - P_X(t)| > \epsilon) \leq 2\exp(-2M\epsilon^2)$. Calculating the Wasserstein distance with the empirical distribution function is computationally attractive. Sorting $x_m$s in an ascending order, such that $x_{i[m]} \leq x_{i[m+1]}$ and where $i[m]$ is the index of the sorted $x_m$s, it is straightforward to see that $P_{X,M}^{-1}(\tau_m) = x_{i[m]}$ (see Fig. 1 for a visualization). The Wasserstein distance can be approximated by first sorting $x_m$s and $y_m$s and then calculating:

$$W_c(p_X, p_Y) \approx W_c(p_{X,M}, p_{Y,M}) = \frac{1}{M} \sum_{m=1}^{M} c(x_{i[m]}, y_{j[m]}) \tag{15}$$

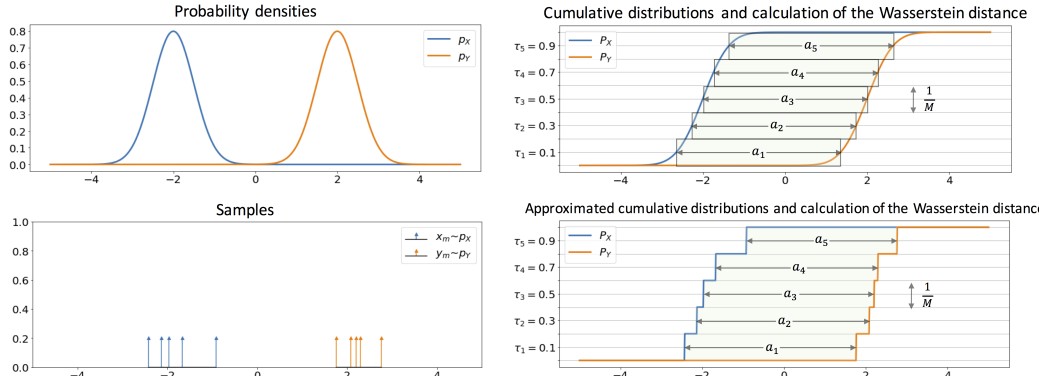

Figure 1: The Wasserstein distance for one-dimensional probability distributions $p_X$ and $p_Y$ (top left) is calculated based on equation 9. For a numerical implementation, the integral in equation 9 is substituted with $\frac{1}{M} \sum_{m=1}^{M} a_m$ where, $a_m = c(P_X^{-1}(\tau_m), P_Y^{-1}(\tau_m))$ (top right). When only samples from the distributions are available $x_n \sim p_X$ and $y_n \sim Y$ (bottom left), the Wasserstein distance is approximated by sorting $x_m$s and $y_m$s and letting $a_m = c(x_{i[m]}, y_{j[m]})$, where $i[m]$ and $j[m]$ are the sorted indices (bottom right).

The problem of calculating the Wasserstein distance between samples from one-dimensional densities simplifies to solving two sorting problems (solved in $\mathcal{O}(M)/\mathcal{O}(Mlog(M))$ best/worst case).

We need to address one final question here. How well does equation 15 approximate the Wasserstein distance, $W_c(p_X, p_Y)$? We first note that the rates of convergence of empirical distributions, for the p-Wasserstein metric (i.e., $c(x, y) = |x - y|^p$) of order $p \geq 1$, have been extensively studied in the mathematics and statistics communities (see for instance Bobkov & Ledoux (2014) and Dedecker et al. (2015)). A detailed description of these rates is, however, beyond the scope of this paper, especially since these rates are dependent on the choice of $p$. In short, for $p = 1$ it can be shown that $\mathbb{E}(W_1(p_{X,M}, p_X) \leq \frac{C}{\sqrt{M}}$ where $C$ is an absolute constant. Similar results are achieved for $\mathbb{E}(W_p(p_{X,M}, p_X))$ and $(\mathbb{E}(W_p^p(p_{X,M}, p_X)))^{\frac{1}{p}}$, although under more strict assumptions on $p_X$ (i.e., slightly stronger assumptions than having a finite second moment). Using the triangle inequality together with the convergence rates of empirical distributions with respect to the p-Wasserstein distance, see Bobkov & Ledoux (2014), for $W_1(p_{X,M}, p_X)$ (or more generally $W_p(p_{X,M}, p_X)$) we can show that (see supplementary material):

$$\mathbb{E}(W_1(p_X, p_Y) - W_1(p_{X,M}, p_{Y,M})) \leq \frac{C}{\sqrt{M}} \tag{16}$$

for some absolute constant, $C$. We reiterate that similar bounds could be found for $W_p$ although with slightly more strict assumptions on $p_X$ and $p_Y$.

## 3.2 SLICING EMPIRICAL DISTRIBUTIONS

In scenarios where only samples from the d-dimensional distribution, $p_X$, are available, $x_m \sim p_X$, the empirical density can be estimated as $p_{X,M} = \frac{1}{M} \sum_{m=1}^{M} \delta_{x_m}$. Following equation 10 it is straightforward to show that the marginal densities (i.e. slices) are obtained from:

$$\mathcal{R}p_X(t, \theta) \approx \mathcal{R}p_{X,M}(t, \theta) = \frac{1}{M} \sum_{m=1}^{M} \delta(t - x_m \cdot \theta), \ \forall \theta \in \mathbb{S}^{d-1}, \text{and} \ \forall t \in \mathbb{R} \tag{17}$$

see the supplementary material for a proof. The Dvoretzky–Kiefer–Wolfowitz upper bound holds for $\mathcal{R}p_X(t, \theta)$ and $\mathcal{R}p_{X,M}(t, \theta)$.

## 3.3 MINIMIZING SLICED-WASSERSTEIN VIA RANDOM SLICING

Minimizing the sliced-Wasserstein distance (i.e., as in the second term of 14) requires an integration over the unit sphere in $\mathbb{R}^d$, i.e., $\mathbb{S}^{d-1}$. In practice, this integration is approximated by using a simple Monte Carlo scheme that draws uniform samples from $\mathbb{S}^{d-1}$ and replaces the integral with a

finite-sample average,

$$SW_c(p_Z, q_Z) \approx \frac{1}{|\Theta|} \sum_{\theta_l \in \Theta} W_c(\mathcal{R}p_Z(\cdot; \theta_l), \mathcal{R}q_Z(\cdot; \theta_l))$$

Such Monte Carlo estimation was used in Rabin & Peyré (2011), and later used in Bonneel et al. (2015); Kolouri et al. (2018); Şimşekli et al. (2018); Deshpande et al. (2018). Moreover, the global minimum for $SW_c(p_Z, q_Z)$ is also a global minimum for each $W_c(\mathcal{R}p_Z(\cdot; \theta_l), \mathcal{R}q_Z(\cdot; \theta_l))$. Note that $SW_c(p_Z, q_Z) = \mathbb{E}_{\mathbb{S}^{(d-1)}}(W_c(\mathcal{R}p_Z(\cdot; \theta), \mathcal{R}q_Z(\cdot; \theta)))$. A fine sampling of $\mathbb{S}^{d-1}$, however, is required for a good approximation of $SW_c(p_Z, q_Z)$. Intuitively, if $p_Z$ and $q_Z$ are similar, then their projections with respect to any finite subset of $\mathbb{S}^{d-1}$ would also be similar. This

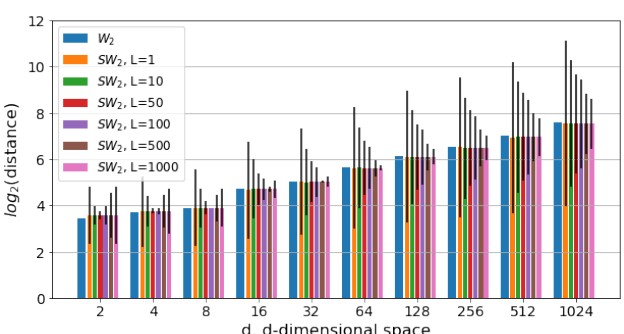

Figure 2: SW approximations (scaled by $1.22\sqrt{d}$) of the W-distance in different dimensions, $d \in \{2^n\}_{n=1}^{10}$, and different number of random slices, $L$.

leads to a stochastic gradient descent scheme where in addition to the random sampling of the input data, we also random sample the projection angles from $\mathbb{S}^{d-1}$.

A natural question arises on the effect of the number of random slices, $L = |\Theta|$, on the approximation of the SW distance. Here, we devised a simple experiment that demonstrates the effect of $L$ on approximating the SW distance. We generated two random multi-variate Gaussian distributions in a $d$-dimensional space, where $d \in \{2^n\}_{n=1}^{10}$, to serve as $p_X = \mathcal{N}(\mu_X, \Sigma_X)$ and $p_X = \mathcal{N}(\mu_Y, \Sigma_Y)$. The Wasserstein distance for the two Gaussian distributions has a closed form solution,

$$W_2^2(p_X, p_Y) = \|\mu_X - \mu_Y\|_2^2 + trace(\Sigma_X + \Sigma_Y - 2(\Sigma_X^{\frac{1}{2}} \Sigma_Y \Sigma_X^{\frac{1}{2}})^{\frac{1}{2}}),$$

which served as the ground-truth distance between the distributions. We then measured the SW distance between $M = 1000$ samples generated from the two Gaussian distributions using $L \in \{1, 10, 50, 100, 500, 1000\}$ random slices. We repeated the experiment for each $L$ and $d$, a thousand times and report the means and standard deviations in Figure 2. Following equation 13 we scaled the SW distance by $\sqrt{d}$. Moreover we found out empirically that $1.22\sqrt{d}\,\mathbb{E}(SW_2(p_{X,M}, p_{Y,M})) \approx W_2(p_X, p_Y)$. It can be seen from Figure 2 that the expected value of the scaled $SW$-distance closely follows the true Wasserstein distance. A more interesting observation is that the variance of estimation increases for higher dimensions $d$ and decreases as the number of random projections, $L$, increases. Hence, calculating the SW distance in the image space, as in Deshpande et al. (2018), requires a very large number of projections $L$ to get a less variant approximation of the distance.

## 3.4 PUTTING IT ALL TOGETHER

To optimize the proposed SWAE objective function in equation 14 we use a stochastic gradient descent scheme as described here. In each iteration, let $\{x_m \sim p_X\}_{m=1}^M$ and $\{\tilde{z}_m \sim q_Z\}_{m=1}^M$ be i.i.d random samples from the input data and the predefined distribution, $q_Z$, correspondingly. Let $\{\theta_l\}_{l=1}^L$ be randomly sampled from a uniform distribution on $\mathbb{S}^{d-1}$. Then using the numerical approximations described in this section, the loss function in equation 14 can be rewritten as:

$$\mathcal{L}(\phi, \psi) = \frac{1}{M} \sum_{m=1}^M c(x_m, \psi(\phi(x_m))) + \frac{\lambda}{LM} \sum_{l=1}^L \sum_{m=1}^M c(\theta_l \cdot \tilde{z}_{i[m]}, \theta_l \cdot \phi(x_{j[m]})) \quad (18)$$

where $i[m]$ and $j[m]$ are the indices of sorted $\theta_l \cdot \tilde{z}_m$s and $\theta_l \cdot \phi(x_m)$ with respect to $m$, correspondingly. The steps of our proposed method are presented in Algorithm 1. It is worth pointing out that sorting is by itself an optimization problem (which can be solved very efficiently), and therefore the sorting followed by the gradient descent update on $\phi$ and $\psi$ is in essence a min-max problem, which is being solved in an alternating fashion. Finally, we point out that each iteration of SWAE costs $\mathcal{O}(LMlog(M))$ operations.

---

**Algorithm 1** Sliced-Wasserstein Auto-Encoder (SWAE)

---

**Require:** Regularization coefficient $\lambda$, and number of random projections, $L$.

   Initialize the parameters of the encoder, $\phi$, and decoder, $\psi$

   **while** $\phi$ and $\psi$ have not converged **do**

      Sample $\{x_1, ..., x_M\}$ from training set (i.e. $p_X$)

      Sample $\{\tilde{z}_1, ..., \tilde{z}_M\}$ from $q_Z$

      Sample $\{\theta_1, ..., \theta_L\}$ from $\mathbb{S}^{K-1}$

      Sort $\theta_l \cdot \tilde{z}_M$ such that $\theta_l \cdot \tilde{z}_{i[m]} \leq \theta_l \cdot \tilde{z}_{i[m+1]}$

      Sort $\theta_l \cdot \phi(x_m)$ such that $\theta_l \cdot \phi(x_{j[m]}) \leq \theta_l \cdot \phi(x_{j[m+1]})$

      Update $\phi$ and $\psi$ by descending: $\sum_{m=1}^{M} c(x_m, \psi(\phi(x_m))) + \lambda \sum_{l=1}^{L} \sum_{m=1}^{M} c(\theta_l \cdot \tilde{z}_{i[m]}, \theta_l \cdot \phi(x_{j[m]}))$

   **end while**

---

## 4 EXPERIMENTS

In our experiments we used three image datasets, namely the MNIST dataset by LeCun (1998), the CelebFaces Attributes Dataset (CelebA) by Liu et al. (2015), and the LSUN Bedroom Dataset by Yu et al. (2015). For the MNIST dataset we used a simple auto-encoder with mirrored classic deep convolutional neural networks with 2D average poolings, leaky rectified linear units (Leaky-ReLu) as the activation functions, and upsampling layers in the decoder. For the CelebA and LSUN datasets we used the DCGAN Radford et al. (2015) architecture similar to Tolstikhin et al. (2017).

To test the capability of our proposed algorithm in shaping the latent space of the encoder, we started with the MNIST dataset and trained SWAE to encode this dataset to a two-dimensional latent space (for the sake of visualization) while enforcing a match between $p_X$ and $p_Y$ and $p_Z$ and $q_Z$. We chose four different samplable distributions as shown in Figure 3. It can bee seen that SWAE can successfully embed the dataset into the latent space while enforcing $p_Z$ to closely follow $q_Z$. In addition, we sample the two-dimensional latent spaces on a $25 \times 25$ grid in $[-1, 1]^2$ and decode these points to visualize their corresponding images in the digit/image space.

To get a sense of the convergence behavior of SWAE, and similar to the work of Karras et al. (2017), we calculate the Sliced Wasserstein distance between $p_Z$ and $q_Z$ as well as $p_X$ and $p_Y$ at each batch iteration where we used p-LDA Wang et al. (2011) to calculate projections (See supplementary material). We compared the convergence behavior of SWAE with the closest related work, WAE Tolstikhin et al. (2017) (specifically WAE-GAN) where an adversarial training is used to match $p_Z$ to $q_Z$, while the loss function for $p_X$ and $p_Y$ remains exactly the same between the two methods. We repeated the experiments 100 times and report the summary of results in Figure 4. We mention that the exact same models and optimizers were used for both methods in this experiment. An interesting observation, here is that while WAE-GAN provides good or even slightly better generated random samples for MNIST (lower sliced-Wasserstein distance between $p_X$ and $p_Y$), it fails to provide a good match between $p_Z$ and $q_Z$ for the choice of the prior distribution reported in Figure 4. This phenomenon seems to be related to the mode-collapse problem of GANs, where the adversary fails to sense that the distribution is not fully covered. Finally, in our experiments we did not notice a significant difference between the computational time for SWAE and WAE-GAN. For the MNIST experiment and on a single NVIDIA Tesla $P100$ GPU, each batch iteration (batchsize=500) of WAE-GAN took $0.2571 \pm 0.0435$(sec) while SWAE (with $L = 50$ projections) took $0.2437 \pm 0.0391$(sec).

$$\psi(t\phi(I_0) + (1-t)\phi(I_1)), \qquad t \in [0,1]$$

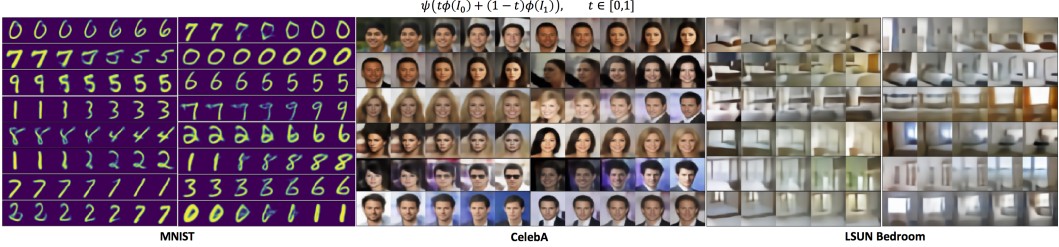

MNIST                 CelebA                 LSUN Bedroom

Figure 5: Interpolation in the latent space, $\psi(t\phi(I_0) + (1-t)\phi(I_1))$ for $t \in [0, 1]$.

| Dataset | Iteration $\cdot 10^{-4}$ | Model | log $\mathrm{SW}(p_z, q_z)$ | log $\mathrm{SW}(p_x, p_y)$ | $\mathrm{NLL}(Z|q_z) \cdot 10^{-4}$ |
|---|---|---|---|---|---|
| CelebA | 1 | SWAE | $-0.81 \pm 0.05$ | $-2.19 \pm 0.04$ | $\mathbf{3.14 \pm 0.05}$ |
| | | WAE-GAN | $-0.78 \pm 0.05$ | $-2.04 \pm 0.05$ | $3.25 \pm 0.15$ |
| | | WAE-MMD(IMQ) | $\mathbf{-1.44 \pm 0.19}$ | $-2.51 \pm 0.05$ | $3.66 \pm 0.12$ |
| | | WAE-MMD(RBF) | $3.26 \pm 0.02$ | $\mathbf{-2.60 \pm 0.02}$ | $2392 \pm 89$ |
| | 5 | SWAE | $-1.80 \pm 0.03$ | $-2.63 \pm 0.03$ | $\mathbf{3.22 \pm 0.02}$ |
| | | WAE-GAN | $-1.37 \pm 0.12$ | $-2.42 \pm 0.05$ | $3.47 \pm 0.13$ |
| | | WAE-MMD(IMQ) | $\mathbf{-2.15 \pm 0.02}$ | $-2.86 \pm 0.01$ | $3.51 \pm 0.04$ |
| | | WAE-MMD(RBF) | $3.28 \pm 0.02$ | $\mathbf{-2.89 \pm 0.02}$ | $2469 \pm 79$ |
| | 10 | SWAE | $-2.01 \pm 0.04$ | $-2.75 \pm 0.03$ | $\mathbf{3.24 \pm 0.00}$ |
| | | WAE-GAN | $\mathbf{-2.33 \pm 0.14}$ | $-2.55 \pm 0.06$ | $3.42 \pm 0.04$ |
| | | WAE-MMD(IMQ) | $-2.23 \pm 0.00$ | $-2.97 \pm 0.01$ | $3.50 \pm 0.01$ |
| | | WAE-MMD(RBF) | $3.23 \pm 0.02$ | $\mathbf{-2.99 \pm 0.02}$ | $2227 \pm 88$ |
| LSUN Bedroom | 1 | SWAE | $-0.98 \pm 0.17$ | $-1.88 \pm 0.06$ | $\mathbf{3.12 \pm 0.07}$ |
| | | WAE-GAN | $-1.18 \pm 0.16$ | $-1.90 \pm 0.07$ | $3.31 \pm 0.16$ |
| | | WAE-MMD(IMQ) | $\mathbf{-1.72 \pm 0.07}$ | $-2.13 \pm 0.02$ | $3.61 \pm 0.04$ |
| | | WAE-MMD(RBF) | $3.45 \pm 0.02$ | $\mathbf{-2.16 \pm 0.04}$ | $3446 \pm 152$ |
| | 5 | SWAE | $-1.94 \pm 0.12$ | $-2.34 \pm 0.04$ | $\mathbf{3.22 \pm 0.02}$ |
| | | WAE-GAN | $\mathbf{-2.34 \pm 0.04}$ | $-2.30 \pm 0.04$ | $3.40 \pm 0.08$ |
| | | WAE-MMD(IMQ) | $-2.21 \pm 0.02$ | $\mathbf{-2.47 \pm 0.02}$ | $3.48 \pm 0.04$ |
| | | WAE-MMD(RBF) | $3.53 \pm 0.03$ | $\mathbf{-2.47 \pm 0.02}$ | $4009 \pm 258$ |
| | 10 | SWAE | $-2.08 \pm 0.11$ | $-2.46 \pm 0.03$ | $\mathbf{3.23 \pm 0.01}$ |
| | | WAE-GAN | $\mathbf{-2.49 \pm 0.02}$ | $-2.41 \pm 0.03$ | $3.35 \pm 0.05$ |
| | | WAE-MMD(IMQ) | $-2.25 \pm 0.02$ | $-2.59 \pm 0.02$ | $3.50 \pm 0.01$ |
| | | WAE-MMD(RBF) | $3.48 \pm 0.04$ | $\mathbf{-2.60 \pm 0.02}$ | $3624 \pm 282$ |

Table 1: Quantitative comparison of the SWAE and WAE-GAN using the sliced-Wasserstein distance with discriminant slices in the latent space, $SW(p_Z, q_Z)$, and the output space, $SW(p_X, p_Y)$. The distribution in the 64-dimensional latent space, $q_Z$, was set to Normal. We also report the negative log-likelihood of $\{z_i = \phi(x_i)\}$ with repect to $q_Z$ for 1000 testing samples for both datasets. We did not use Nowizin's trick for the GAN models.

| Model | FID - CelebA | FID - LSUN Bedroom |
|---|---|---|
| SWAE | $79 \pm 6$ | $\mathbf{225 \pm 7}$ |
| WAE-GAN | $\mathbf{53 \pm 2}$ | $232 \pm 2$ |
| WAE-MMD(IMQ) | $55 \pm 1$ | $226 \pm 2$ |
| WAE-MMD(RBF) | $363 \pm 17$ | $378 \pm 12$ |
| True Data | $2$ | $3$ |

Table 2: FID score statistics ($N = 5$) at final iteration of training. Lower is better. Scores were computed with $10^4$ random samples from the testing set against an equivalent amount of generated samples.

The CelebA face and the LSUN bedroom datasets contain higher degrees of variations compared to the MNIST dataset and therefore a two-dimensional latent-space does not suffice to capture the variations in these datasets (See supplementary material for more details on the dimensionality of the latent space). We used a $K = 64$ dimensional latent spaces for both the CelebA and the LSUN Bedroom datasets, and also used a larger auto-encoder (i.e., DCGAN, following the work of Tolstikhin et al. (2017)). For these datasets SWAE was trained with $q_Z$ being the Normal distribution to enable the calculation of the negative log likelihood (NLL). Table 1 shows the comparison between SWAE and WAE for these two datasets. We note that all experimental parameters were kept the same to enable an apples to apples comparison. Finally, Figure 5 demonstrates the interpolation between two sample points in the latent space, i.e. $\psi(t\phi(I_0) + (1 - t)\phi(I_1))$ for $t \in [0, 1]$, for all three datasets.

## 5 CONCLUSIONS

We introduced Sliced Wasserstein auto-encoders (SWAE), which enable one to shape the distribution of the encoded samples to any samplable distribution without the need for adversarial training or having a likelihood function specified. In addition, we provided a simple and efficient numerical scheme for this problem, which only relies on few inner products and sorting operations in each SGD iteration. We further demonstrated the capability of our method on three image datasets, namely the MNIST, the CelebA face, and the LSUN Bedroom datasets, and showed competitive performance, in the sense of matching distributions $p_Z$ and $q_Z$, to the techniques that rely on additional adversarial trainings. Finally, we envision SWAE could be effectively used in transfer learning and domain adaptation algorithms where $q_Z$ comes from a source domain and the task is to encode the target domain $p_X$ in a latent space such that the distribution follows the distribution of the target domain.

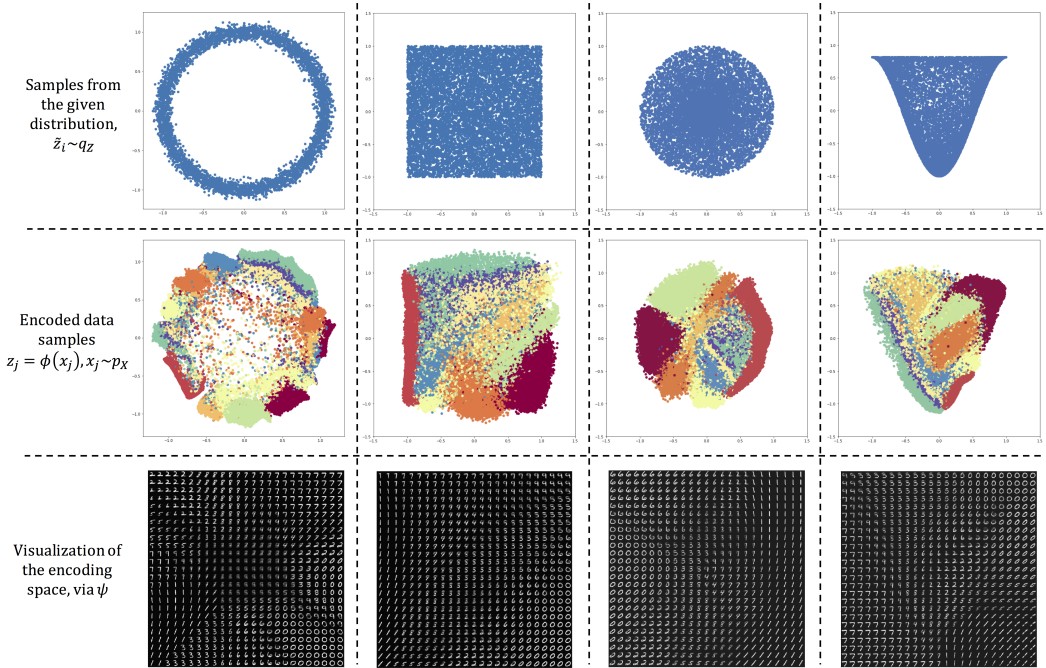

Figure 3: The results of SWAE on the MNIST dataset with a two-dimensional embedding space for four different distributions as ,$q_Z$, namely the ring distribution (top left), the uniform distribution (bottom left), the uniform polar distribution (top right), and a custom polar distribution (bottom right). Note that the far right visualization demonstrates the decoding of a $25 \times 25$ grid in $[-1,1]^2$.

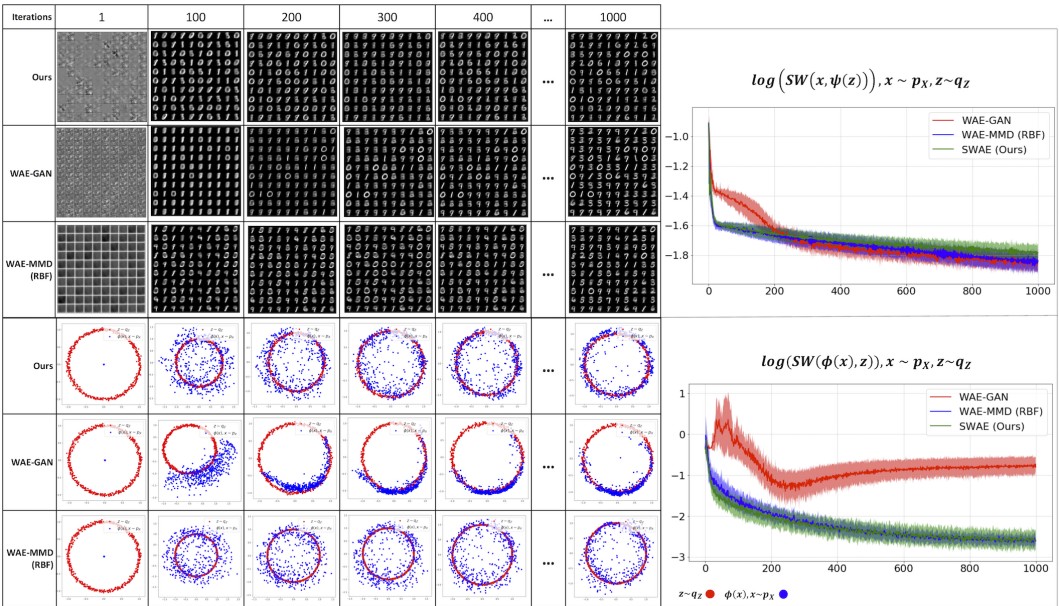

Figure 4: Sample convergence behavior for our method compared to the WAE-GAN, where $q_Z$ is set to a ring distribution (Figure 3, top left). The columns represent batch iterations (batchsize= 500). The top half of the table shows results of $\psi(z)$ for $z \sim q_Z$, and the bottom half shows $z \sim q_Z$ and $\phi(x)$ for $x \sim p_X$. It can be seen that the adversarial loss in the latent space does not provide a full coverage of the distribution, which is a similar problem to the well-known 'mode collapse' problem in the GANs. It can be seen that SWAE provides a superior match between $p_Z$ and $q_Z$ while it does not require adversarial training.

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

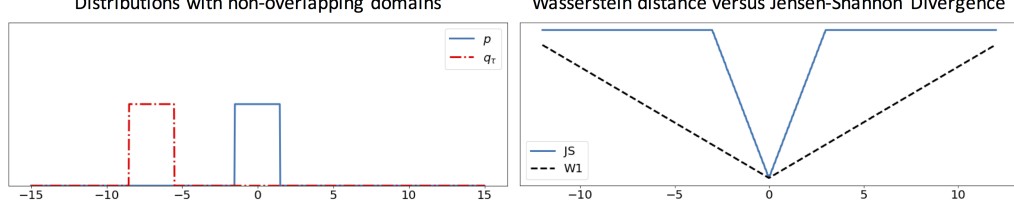

Figure 6: These plots show $W_1(p, q_\tau)$ and $JS(p, q_\tau)$ where $p$ is a uniform distribution around zero and $q_\tau(x) = p(x - \tau)$. It is clear that JS divergence does not provide a usable gradient when distributions are supported on non-overlapping domains.

## SUPPLEMENTARY MATERIAL

### COMPARISON OF DIFFERENT DISTANCES

Following the example by Arjovsky et al. (2017) and later Kolouri et al. (2018) here we show a simple example comparing the Jensen-Shannon divergence with the Wasserstein distance. First note that the Jensen-Shannon divergence is defined as,

$$JS(p, q) = KL(p, \frac{p+q}{2}) + KL(q, \frac{p+q}{2})$$

where $KL(p, q) = \int_X p(x) log(\frac{p(x)}{q(x)}) dx$ is the Kullback-Leibler divergence. Now consider the following densities, $p(x)$ be a uniform distribution around zero and let $q_\tau(x) = p(x - \tau)$ be a shifted version of the $p$. Figure 6 show $W_1(p, q_\tau)$ and $JS(p, q_\tau)$ as a function of $\tau$. As can be seen the JS divergence fails to provide a useful gradient when the distributions are supported on non-overlapping domains.

### LOG-LIKELIHOOD

To maximize (minimize) the similarity (dissimilarity) between $p_Z$ and $q_Z$, we can write :

$$\text{argmax}_\phi \int_Z p_Z(z) log(q_Z(z)) dz = \int_Z \int_X p_X(x) \delta(z - \phi(x)) log(q_Z(z)) dx dz$$
$$= \int_X p_X(x) log(q_Z(\phi(x))) dx$$

where we replaced $p_Z$ with equation 1. Furthermore, it is straightforward to show:

$$\text{argmax}_\phi \int_Z p_Z(z) log(q_Z(z)) dz = \text{argmax}_\phi \int_Z p_Z(z) log(\frac{q_Z(z)}{p_Z(z)}) dz$$
$$= \text{argmin}_\phi D_{KL}(p_Z, q_Z)$$

### PROOF OF EQUATION 3

The Wasserstein distance between the two probability measures $\rho_X$ and $\rho_Y$ with respective densities $p_X$ and $p_Y$, can be measured via the Kantorovich formulation of the optimal mass transport problem:

$$W_c(p_X, p_Y) = \inf_{\gamma \in \Gamma} \int_X \int_Y c(x, y) \gamma(x, y) dx dy$$

where $\Gamma := \{\gamma : X \times Y \to \mathbb{R}^+ | \int_Y \gamma(x, y) dy = p_X(x), \int_X \gamma(x, y) dx = p_Y(y)\}$ is the set of all transportation plans (i.e., couplings or joint distributions) over $p_X$ and $p_Y$. Now, note that the two step process of encoding $p_X$ into the latent space $Z$ and decoding it to $p_Y$, provides a unique decomposition of $\gamma$ as $\gamma_0(x, y) = \delta(y - \psi(\phi(x))) p_X(x) \in \Gamma$.

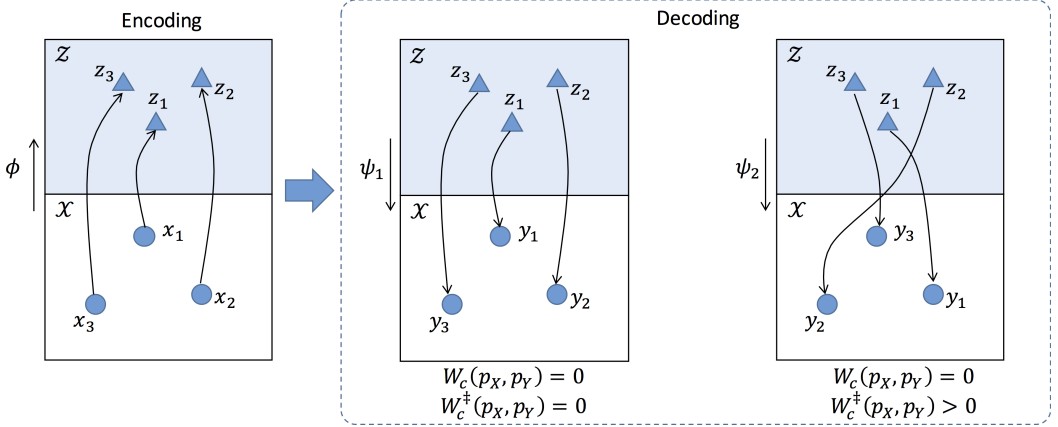

Figure 7: The optimal coupling (i.e., transport plan) between $p_X$ and $p_Y$ could be equal or different from $\gamma(x, y) = \delta(y - \psi(\phi(x)))p_X(x)$. This leads to the scenario on the right where $W_c(p_X, p_Y) = 0$ but $W_c^\ddagger(p_X, p_Y) > 0$.

Therefore we can write:

$$W_c(p_X, p_Y) = \inf_{\gamma \in \Gamma} \int_X \int_Y c(x, y)\gamma(x, y)dxdy \le$$

$$W_c^\ddagger(p_X, p_Y) := \int_X \int_Y c(x, y)\gamma_0(x, y)dxdy = \int_X c(x, \psi(\phi(x)))p_X(x)dx$$

which proves equation 3. Finally, taking the infimum of the two sides of the inequality, with respect to $\phi$ and $\psi$, we have:

$$\inf_{\psi, \phi} W_c(p_X, p_Y) = \inf_{\psi, \phi} \inf_{\gamma \in \Gamma_{\psi, \phi}} \int_X \int_Y c(x, y)\gamma(x, y)dxdy \le$$

$$\inf_{\psi, \phi} W_c^\ddagger(p_X, p_Y) = \inf_{\psi, \phi} \int_X c(x, \psi(\phi(x)))p_X(x)dx$$

where $\Gamma_{\psi, \phi} := \{\gamma| \int_Y \gamma(x, y)dy = p_X(x), \int_X \gamma(x, y)dx = \int_X p_X(x)\delta(y - \psi(\phi(x)))dx\}$. Figure 7 demonstrates a simple scenario were the Wasserstein distance, $W_c(p_X, p_Y)$, is zero however, $W_c^\ddagger(p_X, p_Y)$ is non-zero. Finally, we note that $\psi(\phi(\cdot)) = id(\cdot)$ is a global optima for both $W_c(p_X, p_Y)$ and $W_c^\ddagger(p_X, p_Y)$.

### SLICING EMPIRICAL DISTRIBUTIONS

Following equation 10 a distribution can be sliced via:

$$\mathcal{R}p_X(t, \theta) = \int_X p_X(x)\delta(t - \theta \cdot x)dx$$

Figure 8 visualizes two sample slices for an example distribution $p_X$. Here we calculate a Radon slice of the empirical distribution $p_X(x) = \frac{1}{M}\sum_{m=1}^{M} \delta(x - x_m)$ with respect to $\theta \in \mathbb{S}^{d-1}$. Using the definition of the Radon transform in equation 10 and RVT in equation 1 we have:

$$\mathcal{R}p_X(t, \theta) = \frac{1}{M}\sum_{m=1}^{M} \int_X \delta(x - x_m)\delta(t - \theta \cdot x)dx$$

$$= \frac{1}{M}\sum_{m=1}^{M} \delta(t - \theta \cdot x_m)$$

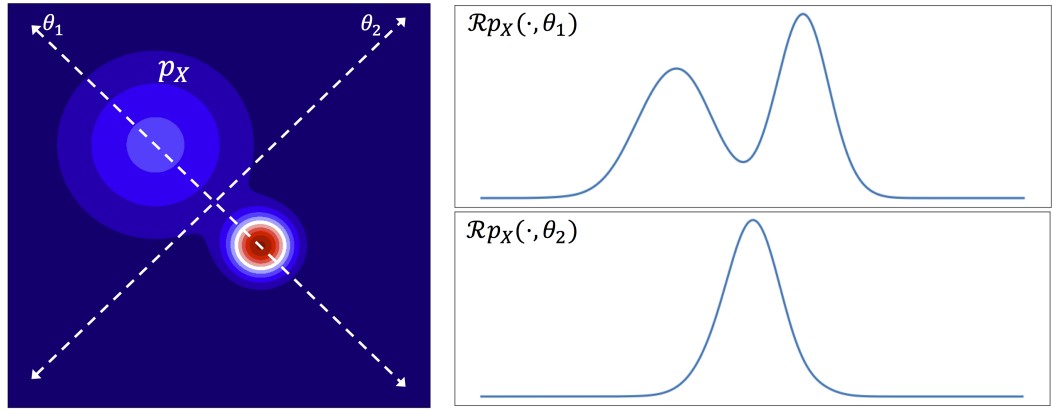

Figure 8: Visualization of the slicing process defined in equation 10

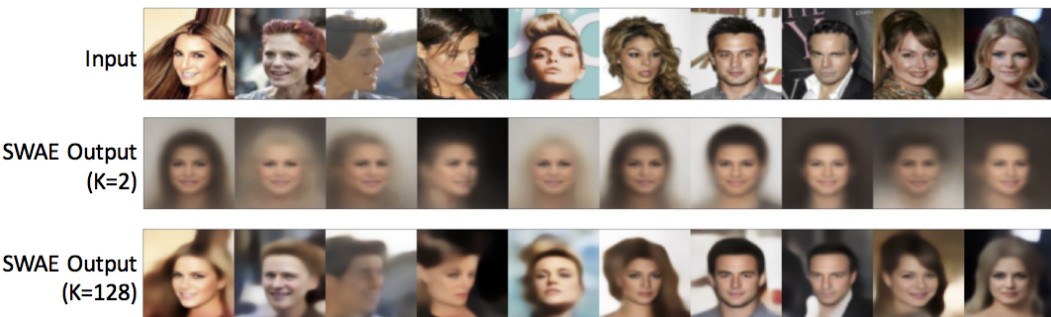

Figure 9: Trained SWAE outputs for sample input images with different embedding spaces of size $K = 2$ and $K = 128$.

### DIMENSIONALITY OF THE LATENT SPACE

Figure 9 demonstrates the outputs of trained SWAEs with $K = 2$ and $K = 128$ for sample input images. The input images were resized to $64 \times 64$ and then fed to our auto-encoder structure. This effect can also be seen for the MNIST dataset as shown in Figure 10. When the dimensionality of the latent-space (i.e. information bottleneck) is too low the latent space will not contain enough information to reconstruct crisp images. Increasing the dimensionality of the latent space leads to crisper images.

### CALCULATING THE SLICED WASSERSTEIN DISTANCE AS A MEASURE OF GOODNESS OF FIT

In this paper we also used the sliced Wasserstein distance as a measure of goodness of fit (for convergence analysis). To provide a fair comparison between different methods, we avoided random projections for this comparison. Instead, we calculated a discriminant subspace to separate $\psi(z)$ from $\psi(\phi(x))$ for $z \sim q_Z$ and $x \sim p_X$, and set the projection parameters $\theta$s to the calculated discriminant components. This will lead to only slices that contain discriminant information. We point out that the linear discriminant analysis (LDA) is not a good choice for this task as it only leads to one discriminant component (because we only have two classes). We used the penalized linear discriminant analysis (p-LDA) that utilizes a combination of LDA and PCA. In short, p-LDA solves the following objective function:

$$\operatorname{argmax}_\theta \quad \frac{\theta^T S_T \theta}{\theta^T (S_W + \alpha I) \theta}$$
$$s.t. \quad \|\theta\| = 1$$

Interpolation in Image Space  Interpolation in the embedding space

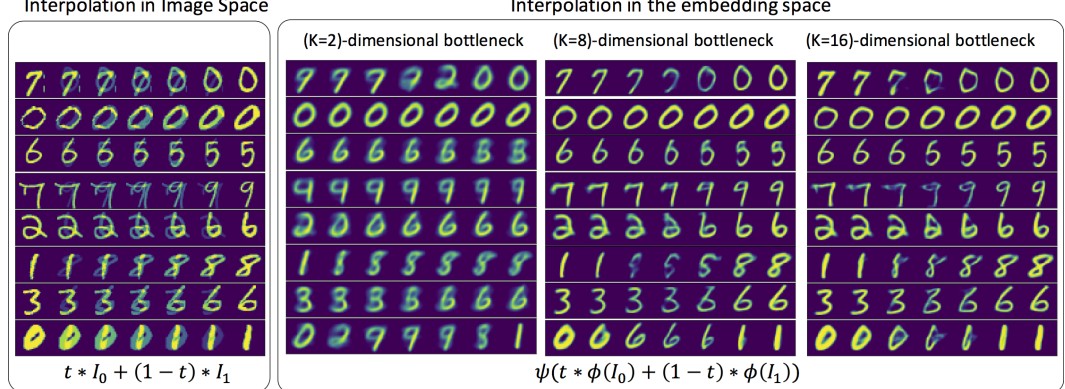

Figure 10: Interpolation results for on the MNIST dataset with various dimensions of the latent space. The parameter $t \in [0, 1]$ indicates the interpolation parameter.

where $S_W$ is the within class covariance matrix, $S_T$ is the data covariance matrix, $I$ is the identity matrix, and $\alpha$ identifies the interpolation between PCA and LDA (i.e. $\alpha = 0$ leads to LDA and $\alpha \rightarrow \infty$ leads to PCA).

ERROR ANALYSIS OF WASSERSTEIN DISTANCE

For $p \geq 1$ we can use the triangle inequality and write

$$
\begin{aligned}
W_p(p_X, p_Y) &\leq W_p(p_X, p_{X,M}) + W_p(p_Y, p_{X,M}) \\
&\leq W_p(p_X, p_{X,M}) + W_p(p_Y, p_{Y,M}) + W_p(p_{X,M}, p_{Y,M})
\end{aligned}
$$

which leads to

$$
W_p(p_X, p_Y) - W_p(p_{X,M}, p_{Y,M}) \leq W_p(p_X, p_{X,M}) + W_p(p_Y, p_{Y,M})
$$

Taking the expectation of both sides of the inequality and using the empirical convergence bounds of $W_p$ (in this case $W_1$) we have,

$$
\begin{aligned}
\mathbb{E}(W_1(p_X, p_Y) - W_1(p_{X,M}, p_{Y,M})) &\leq \mathbb{E}(W_1(p_X, p_{X,M})) + \mathbb{E}(W_1(p_Y, p_{Y,M})) \\
&\leq \frac{C}{\sqrt{M}}
\end{aligned}
$$

for some absolute constant $C$, where the last line comes from the empirical convergence bounds of distributions with respect to the Wasserstein distance, see Bobkov & Ledoux (2014).

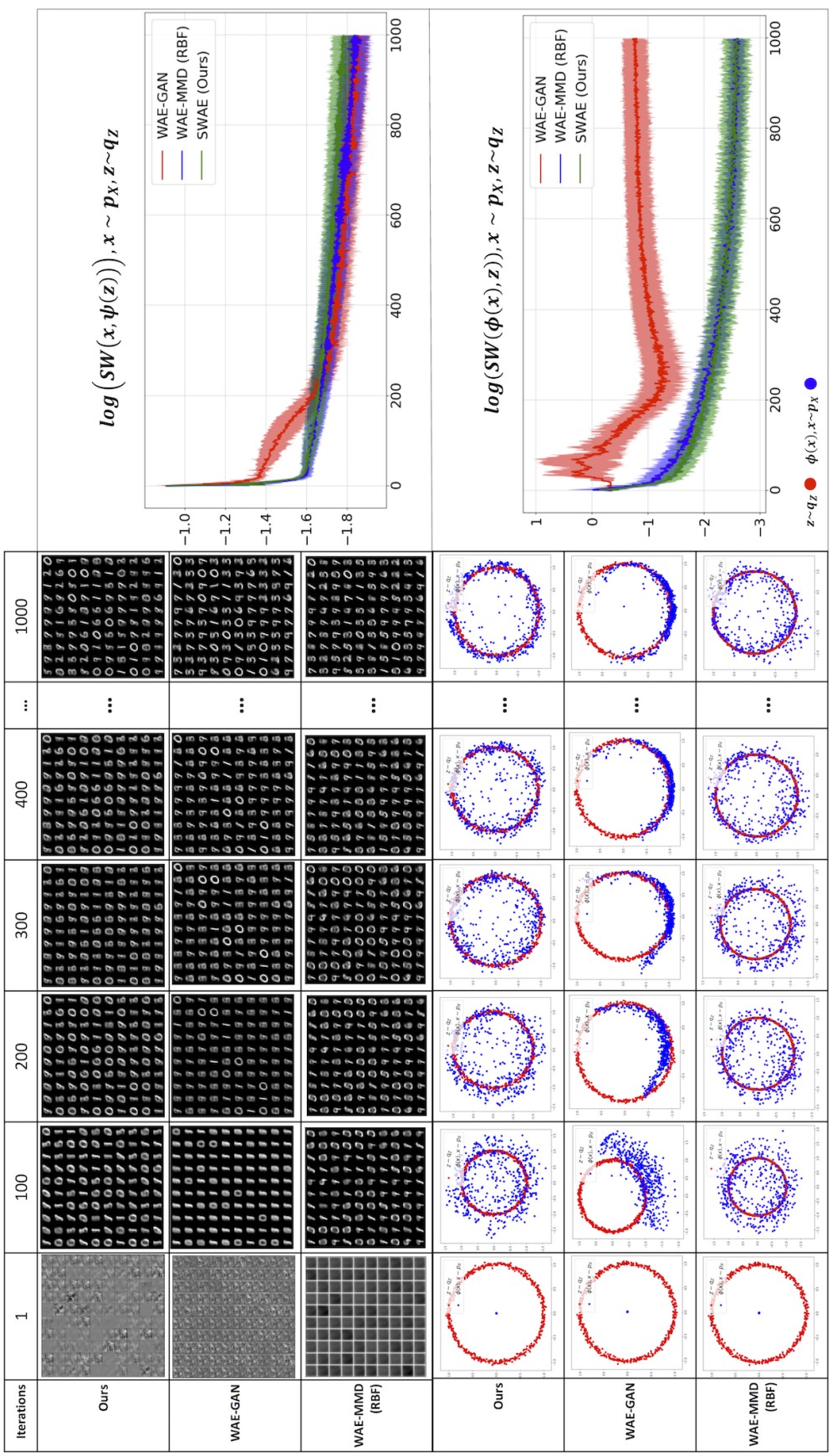

Figure 11: Sample convergence behavior for our method compared to the WAE-GAN Tolstikhin et al. (2017), where $q_Z$ is set to the ring distribution. The columns represent batch iterations (batchsize= 500). The top half of the table shows results of $\psi(z)$ for $z \sim q_Z$, and the bottom half shows results $z \sim q_Z$ and $\phi(x)$ for $x \sim p_X$. It can be seen that SWAE provides a superior match between $\phi(x)$ (i.e. $p_Z$) and $q_Z$ while being less computationally expensive and not requiring adversarial training.

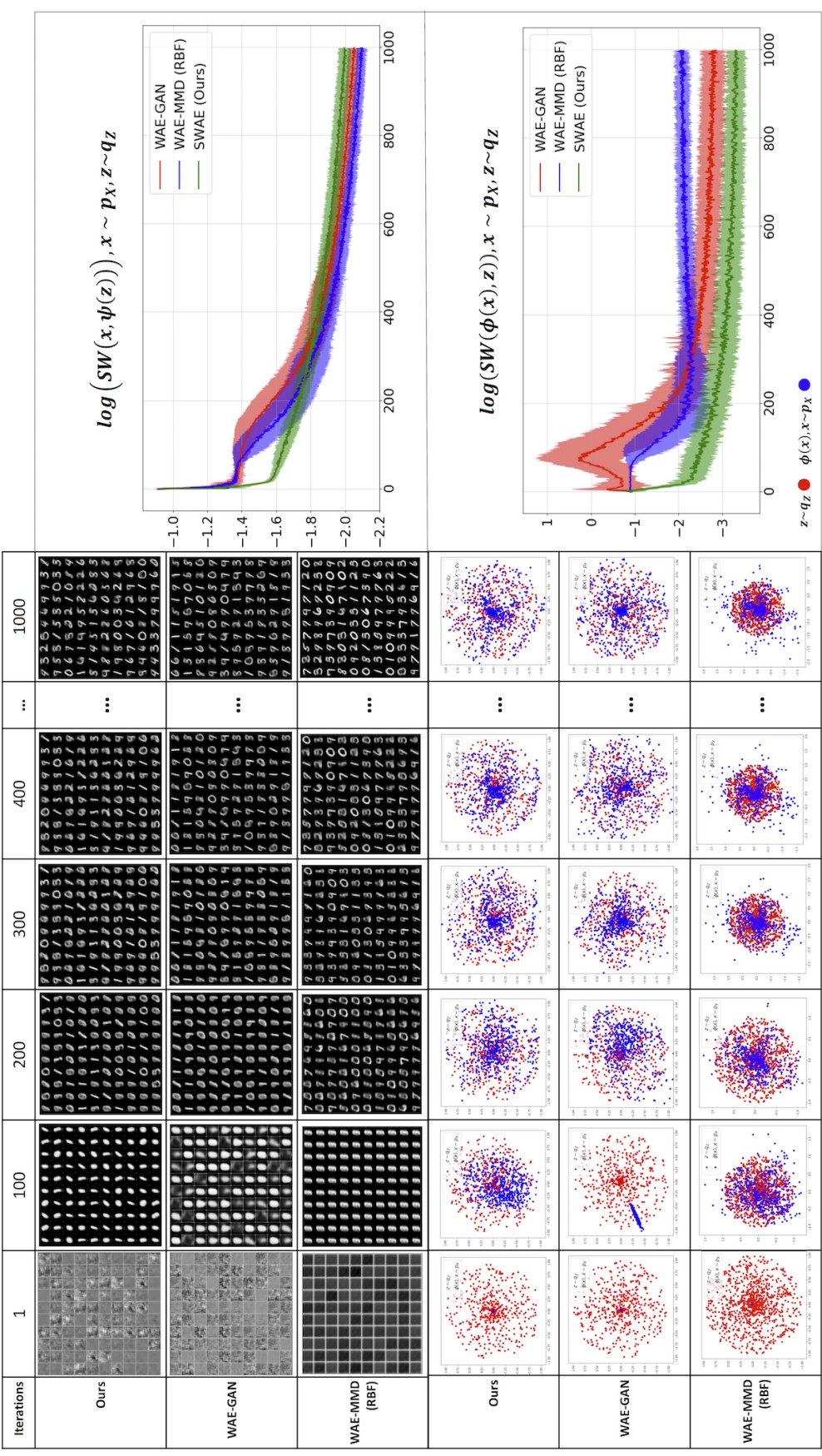

Figure 12: Sample convergence behavior for our method compared to the WAE-GAN Tolstikhin et al. (2017), where $q_Z$ is set to the uniform polar distribution. The columns represent batch iterations (batchsize= 500). The top half of the table shows results of $\psi(z)$ for $z \sim q_Z$, and the bottom half shows $z \sim q_Z$ and $\phi(x)$ for $x \sim p_X$. It can be seen that SWAE provides a superior match between $\phi(x)$ (i.e. $p_Z$) and $q_Z$ while being less computationally expensive and not requiring adversarial training.

