# OpenReview forum: "Sliced Wasserstein Auto-Encoders"
_ICLR.cc/2019/Conference_

### Official Review · AnonReviewer3 · 2018-11-01
**An interesting work which adopts the sliced-Wasserstein distance to simplify the realization of Wasserstein autoencoder**

**Rating:** 6
**Confidence:** 4

**Review:**

This paper presents an extension of Wasserstein autoencoder (WAE) by modifying the regularization term in learning objective of variational autoencoder. This term measures the divergence between the distribution of the encoded training samples and the samplable prior distribution. The modification is based on the sliced-Wasserstein distance where the distance between two distributions is measured through slicing or projecting the high-dimensional distributions into one-dimensional marginal distributions. As a result, a closed-form solution to the integral in Eq. (9) is obtained via a numerical method. The adversarial learning in WAE, designed to fulfill the calculation of high-dimensional distance, can be avoided. In general, this is an interesting work by introducing new idea of sliced-Wasserstein distance.

Remarks:
1. A theoretical paper which addresses how and why the sliced-Wasserstein distance between p_z and q_z is reasonable to build a new variant of variational auto-encoder.
2.  Reformulating the Wasserstein distance into Monge primal formulation with the assumption based on the property of diffeomorphic mapping.
3. As a result, the implementation based on the unstable adversarial training or the maximum mean discrepancy (MMD) training can be avoided. Computational attractiveness is assured. MMD needs the choice of kernel function which is basically a data-dependent design parameter.
4. Provide an empirical numerical solution which is compatible with SGD optimization.
5. The key idea of this paper is shown in Eq. (14). Learning objective is expressed in a deterministic way. However, the style of objective in Eq. (14) involves the stochastic learning.
6. This paper is not actually doubly-blind reviewed. Authors have exposed their identities in arXiv.

---

> ### Author Response · Authors · 2018-11-26
> **Response to Reviewer 3**
>
> We thank the reviewer for a positive evaluation of our work and the constructive feedback.
>
> As precisely pointed out by the reviewer, our paper provides:
> 1.         The theoretical grounds for using sliced-Wasserstein distance as a metric for distributions in deep learning applications
> 2.         Avoiding adversarial training (as in GANs) or the choice of an appropriate kernel with its corresponding data-dependent parameter (as in MMDs)
> 3.         Providing a numerical solution, which is compatible with SGD optimization, and a thorough error analysis of this numerical solution
>
> Regarding the double-blind comment, the ICLR’19 website specifically states that:
>
> “Submissions that are identical (or substantially similar) to versions that have been previously published, or accepted for publication, or that have been submitted in parallel to other conferences or journals, are not allowed and violate our dual submission policy.  However, papers that cite previous related work by the authors and papers that have appeared on non-peered reviewed websites (like arXiv) or that have been presented at workshops (i.e., venues that do not have a publication proceedings) do not violate the policy. The policy is enforced during the whole reviewing process period.”
>
> We would also like to point out that the arxiv submission is significantly different from our ICLR submission as it does not contain:
>
> 1.           Error analysis on our numerical method
> 2.           Quantitative performance measures on the three datasets
>
> and it lacks the theoretical discussion presented in the current submission.
>
> Once again, we thank the reviewer for a precise evaluation of our work and recognizing the novelties of the proposed framework.

---

> > ### Author Response · Authors · 2018-11-30
> > **Further Update**
> >
> > We also just became aware of another submission to ICLR2019, “Cramer-Wold AutoEncoder” (CWAE) https://openreview.net/forum?id=rkgwuiA9F7 , in which the authors cite our work and provide an extensive comparison of our method (SWAE) to WAE and their proposed framework (CWAE). This comparison was possible as we released our code in April 2018. The authors of CWAE kindly provide further quantitative results (See Figure 4 of their paper) for SWAE and WAE on CelebA (and for CIFAR10 in the supplementary material) comparing the FID score, and Mardia’s skewness and kurtosis of models WAE, SWAE and CWAE, on the CelebA test set. CWAE as the authors put it “can be seen as a borderline model between SWAE and WAE-MMD”, “which has a closed-form for the distance of a sample from standard multivariate normal distribution.”
> >
> > We hope that the extensive analysis provided by the authors of CWAE, in addition to our extended experimental section and numerical analysis, to further convince the reviewer on the merit of our proposed method.

---

### Official Review · AnonReviewer1 · 2018-11-01

**Rating:** 4
**Confidence:** 4

**Review:**

This paper proposes training generative models with Wasserstein auto-encoders. It uses the sliced-Wasserstein distance to measure the dissimilarity between p_z and q_z.

Strengths:
1.    This paper is easy to read.
2.    Concepts are introduced clearly.

My major comments are the following:
1.  The innovation is a bit on the incremental level, especially given the results from WAE (Tolstikhin, ICLR18). The training objective is the same as Eq(4) in the WAE paper. The only difference is that the dissimilarity measure between p_z and q_z used in this paper is the sliced- Wasserstein distance, while WAE used GAN/MMD-based penalties. The advantage of using sliced-Wasserstein distance is not clear to me either.

2. The empirical results are fairly weak.  The authors may consider reporting the sample qualities (e.g. FID) for all the methods.

3. The results of WAE-MMD are not reported.

---

> ### Author Response · Authors · 2018-11-26
> **Response to Reviewer 1**
>
> We would like to thank the reviewer for the thorough evaluation of our paper.
>
> ==> The innovation is a bit on the incremental level
>
> Regarding the innovation comment, here we reiterate our specific contributions in this paper.
> 1.         We introduce the sliced-Wasserstein distance (SWD) as a measure for minimizing the difference between distributions in the latent space of an auto-encoder. Our proposed method:
> a.         Does not require adversarial training as in WAE-GAN
> b.         Does a better job at matching the latent distribution of the input data to the prior distribution (See Figures 3 and 12) on MNIST, and provides consistently high performance on CelebA and LSUN Bedroom datasets, and has the best likelihood scores.
> c.          Does not require a choice of kernel and the corresponding kernel parameters (e.g., spread in RBF) as in WAE-MMD.
> 2.         We provide a thorough numerical section with error analysis that supports the effectiveness of the sliced-Wasserstein distance. Through empirical results, we demonstrate that SWD is a good approximation to the true Wasserstein distance.
> 3.         The proposed method has a very simple, yet elegant, numerical implementation, and provides a differentiable loss function, thereby permitting the application of stochastic gradient descent.
>
>
> ==> The empirical results are fairly weak and the results of WAE-MMD are not reported.
>
> We agree with the reviewer that the submitted manuscript lacked extensive quantitative results. We have updated our experimental section to provide more quantitative analysis. Specifically,
>
> 1.         We have added comparison with WAE-MMD for all datasets
> 2.         We have updated Table 1 to report results (i.e., SWD(p_Z,q_Z), SWD(p_X,q_X), and NLL(p(z|q_Z))) for SWAE, WAE-GAN, WAE-MMD (IMQ), and WAE-MMD (RBF)
> 3.         We have reported the FID scores for the CelebA and LSUN datasets
>
> With regards to the use of MMD as a measure of discrepancy between the latent distributions, we would like to point out that:
> 1.         MMD is sensitive to the choice of the kernel (See for example Tolstikhin et al. 2017, in which the authors say: “We tried WAE-MMD  with  the  RBF  kernel  but observed that it fails to penalize the outliers of $Q_Z$ because of the quick tail decay. If the codes $\tilde{z} = \mu_{phi}(x)$ for some of the training points $x \in \mathcal{X}$ end up far away from the support pf $P_Z$ (which may happen in the early stages of training) the corresponding terms in the U-statistic [i.e. the RBF kernel] will quickly approach zero and provide no gradient for those outliers.”)
> 2.         MMD’s computational complexity for each iteration is $\mathcal{O}(N^2)$ where $N$ is the batchsize (as opposed to $\mathcal{O}(Nlog(N))$ of SWAE in the worst case scenario)
> 3.         MMD has additional kernel parameters (e.g., the spread of the RBF signal) to be tuned for the dataset that could significantly affect the performance of the system.
>
> We would like to thank the reviewer for a thorough assessment of our work and hope that we have addressed the concerns.

---

> > ### Author Response · Authors · 2018-11-30
> > **Further update**
> >
> > We also just became aware of another submission to ICLR2019, “Cramer-Wold AutoEncoder” (CWAE) https://openreview.net/forum?id=rkgwuiA9F7 , in which the authors cite our work and provide an extensive comparison of our method (SWAE) to WAE and their proposed framework (CWAE). This comparison was possible as we released our code in April 2018. The authors of CWAE kindly provide further quantitative results (See Figure 4 of their paper) for SWAE and WAE on CelebA (and for CIFAR10 in the supplementary material) comparing the FID score, and Mardia’s skewness and kurtosis of models WAE, SWAE and CWAE, on the CelebA test set. CWAE as the authors put it “can be seen as a borderline model between SWAE and WAE-MMD”, “which has a closed-form for the distance of a sample from standard multivariate normal distribution.”
> >
> > We hope that the extensive analysis provided by the authors of CWAE, in addition to our extended experimental section and numerical analysis, to further convince the reviewer on the merit of our proposed method.

---

### Official Review · AnonReviewer2 · 2018-11-02
**The paper proposes a new auto-encoder model, but possibly contains errors in math and the experiments are not convincing enough.**

**Rating:** 6
**Confidence:** 4

**Review:**

The authors propose a new autoencoding algorithm for the unsupervised generative modeling which they call Sliced Wasserstein Autoencoders (SWAE). SWAE minimizes a reconstruction cost (measured with respect to the non-negative cost function c(x,x') defined for pairs of input images x, x'), regularized by a penalty measuring a discrepancy between the prior distribution over the latent space qz and the push-forward pz of the unknown data distribution through the deterministic encoder. The authors present an extensive theoretical argument supporting the choice of this objective and a number of empirical results performed on MNIST, LSUN bedrooms, and Celeba.

Even though this paper raises several interesting questions, I have several major issues with it:
****
**** 1. Claim around Equation 3 is not proved.
****
All the sections before 2.3 are providing a rather detailed theoretical argument meant to support the choice of the SWAE objective appearing in Eq. 14 of Section 2.3. Here I wand to point out to a mathematical inaccuracy in the authors' discussions, which may render the whole argument questionable. In short, the authors claim around Eq. 3 that "Eq. 3 is equivalent to Theorem 1 of [1] for deterministic encoder-decoder pairs" and don't provide any proofs for this nontrivial fact.

The following is based on some quick derivations I did while reviewing.

Recall that in the current paper Px is the data distribution, Py is the push-forward of Px through the superposition of the encoder \phi and decoder \psi (in other words Py is a distribution of \psi(\phi(X)) when X is distributed according Px). The authors state that:
   \inf_{\phi, \psi} Wc(Px, Py)
    is equivalent to
   (* ) \inf_{\phi, \psi} E_{X \sim Px}[ c(X, \psi(\phi(X))) ].
In other words, the authors state that using Theorem 1 of [1] they are able to show that minimizing a c-optimal transport distance between Px and Py (which is parametrized by \psi and \phi) is *equivalent* to an unconstrained optimization problem appearing on the r.h.s. of Equation 3.

Now, the Theorem 1 of [1] referenced by authors states that if Pz is any prior distribution over the latent space and \psi * Pz is its push-forward through the deterministic decoder \psi, then the optimal transport between Px and the resulting latent variable model \psi * Pz can be equivalently written as:
   (**) Wc(Px, \psi * Pz) = \inf_{f such that f * Px = Pz} E_{X \sim Px}[ c(X, \psi(f(X))) ].
Importantly, note how the right hand side of (**) contains a constrained optimization over an auxiliary (encoder) function f, which does not appear at all in the left hand side. If the authors were to apply (**) directly, they would arrive at the following statement:
   \inf_{\phi, \psi} Wc(Px, Py)
    is equivalent to
   (*** ) \inf_{\phi, \psi} \inf_{f such that f * Px = \phi * Px} E_{X \sim Px}[ c(X, \psi(f(X))) ].
Finally, comparing (*) stated by the authors and (***) obtained above, we see that (*) is obtained by selecting one particular function f = \phi from the set {f such that f * Px = \phi * Px}. Meanwhile, this set in general may contain multiple other functions f and as a result this only shows that (*) >= (***) (as we replace \inf_f with one particular choice of f). However, in this case, I think it is indeed possible to show that (*) = (***). Imagine (***) has a global minimum at (\psi_0,\phi_0, f_0), that is the global optimum of (***) equals E_{X \sim Px}[ c(X, \psi_0(f_0(X))) ]. The same value can be achieved by (*) by setting \phi = f_0. QED.

Once again, these are my preliminary derivations and they need to be checked. But it looks like the claim of the authors is indeed true.

****
**** 2. Empirical evidence is not convincing. ****
****
The main topic of the paper is the unsupervised generative modeling, and the authors claim certain improvements in this field compared to the previous literature. Even though there are no ultimate evaluation metrics available in the field, recently the researchers started supporting their methods with several metrics, including FID scores. By now for most of the widely used datasets the state of art FID scores are well known. In all the experiments the authors provide pictures and interpolations (last row of Fig. 3, Fig. 5) without numbers. I would say nowadays presenting pictures is not enough (being too subjective) and at least some objective numbers (preferably FID) capturing the quality of generated samples should be reported. The authors go into detailed measurements of discrepancy between the aggregate posterior pz and the prior qz, but it is not clear how this affects the actual sample generation. Finally, it is not clear why the authors compare only to WAE-GAN and did not consider WAE-MMD, which is free of adversarial training (in contrast to WAE-GAN) and thus has a stable training and does not involve extra computations of updating the discriminator (as noted by authors on page 10).

[1] Bousquet et al., 2017.

---

> ### Author Response · Authors · 2018-11-26
> **Response to Reviewer 2**
>
> We sincerely thank the reviewer for the exemplar review, and appreciate the depth of the provided feedback. Both points raised by the reviewer are valid and precise, and we agree with them.
>
> **Theory** Regarding the theoretical discrepancy around Equation (3), the concern comes from the fact that calculation of the Wasserstein distance (for d>1) requires solving an optimization to find an optimal coupling (i.e., transport plan as in the Kantorovich’s formulation) or an optimal transport map (from Monge’s formulation of the problem). Therefore, minimizing the Wasserstein distance between p_X and p_Y, with respect to encoder and decoder, \phi and \psi, should also contain a minimization over the set of transport plans or the transport map. This leads to the reviewer’s point that W_c(p_X,p_Y) \leq E_{x \sim p_X} c(x,\psi(\phi(x)). The r.h.s. of the inequality is the transport cost (using the parlance of optimal mass transportation) with respect to the transport plan \gamma(x,y)=\delta(y-\psi(\phi(x)))p_X(x) (induced by the encoder and decoder) which is not necessarily the optimal transport plan between p_X and p_Y, and hence the r.h.s. is greater or equal to the Wasserstein distance (i.e., the optimal transportation cost).
>
> This was an oversight and is corrected in the updated manuscript. In addition, we have added an extensive theoretical discussion on this matter to the supplementary material. We, however, emphasize that our main contribution is on measuring the discrepancy between p_Z and q_Z and therefore the corrections with respect to Equation (3) do not influence the main message of the paper.
>
> **Empirical evidence** We agree with the reviewer that the submitted manuscript lacked more quantitative comparison like the FID scores for the generated samples and have updated our experimental section to provide more quantitative analysis. Specifically,
>
> 1.         We have added comparison with WAE-MMD for all datasets
> 2.         We have updated Table 1 to report results (i.e., SWD(p_Z,q_Z), SWD(p_X,q_X), and NLL(p(z|q_Z))) for SWAE, WAE-GAN, WAE-MMD (IMQ), and WAE-MMD (RBF)
> 3.         We have reported the FID scores for the CelebA and LSUN datasets
>
> However, we emphasize that the main point of our paper is not “unsupervised generative modeling”, but rather having control over the distribution of the embedded data in the latent space. The discrepancy between p_Z and q_Z is crucial to many applications, including transfer learning and domain adaptation, but does not necessarily increase the generated sample qualities. In fact, the updated Figures 3 and 12 indicate that creating a better match between p_Z and q_Z can, in some cases, impose too strong a constraint on the decoder, which reduces the match between p_X and p_Y (i.e., results in lower quality generated samples).
>
> Finally, we have added the comparison with WAE-MMD, which is a kernel-based method. MMD is an effective way of measuring distributional discrepancy between p_Z and q_Z, however, it has the following downsides:
> 1.         It is sensitive to the choice of the kernel (See for example Tolstikhin et al. 2017, in which the authors say: “We tried  WAE-MMD  with  the  RBF  kernel  but observed that it fails to penalize the outliers of $Q_Z$ because of the quick tail decay. If the codes $\tilde{z} = \mu_{phi}(x)$ for some of the training points $x \in \mathcal{X}$ end up far away from the support of $P_Z$ (which may happen in the early stages of training) the corresponding terms in the U-statistic [i.e. the RBF kernel] will quickly approach zero and provide no gradient for those outliers.”)
> 2.         MMD’s computational complexity for each iteration is $\mathcal{O}(N^2)$ where $N$ is the batchsize (as opposed to $\mathcal{O}(Nlog(N))$ of SWAE in the worst case scenario)
> 3.         MMD has additional kernel parameters (e.g., the spread of the RBF signal) to be tuned for the dataset that could significantly affect the performance of the system. In addition, the IMQ kernel implicitly requires knowledge of the latent distribution q_Z (through the C parameter), whereas our approach does not.
>
> Again, we would like to thank the reviewer for the very precise evaluation of our work and setting a high bar for the reviewing process.

---

> > ### Comment · AnonReviewer2 · 2018-11-29
> > **Raising my score**
> >
> > I thank the authors for their reply.
> >
> > After giving it a thought, I indeed agree that the issue with one of the proofs I mentioned earlier does not indeed affect the main message of the paper (and moreover now seems to be completely fixed).
> >
> > Couple of comments on the Table 1 and Table 2:
> > (1) What is Nowozin's trick? Is it important to mention it in the table legend?
> > (2) In table 2 the FID score of WAE-GAN is reported to be 53. The authors also say explicitly that they use architectures similar to the "Wasserstein Auto-Encoders" paper, which reports score 42 for WAE-GAN trained on the same CelebA dataset. Is there a confusion?
> > (3) How exactly do you evaluate the log likelihood in Table 1? If I am not mistaken, you are trying to evaluate either E_qz[ log pz(Z) ] or E_pz[ log qz(Z) ]. Even if qz is Gaussian, pz is intractable.

---

> > > ### Author Response · Authors · 2018-11-30
> > > **Thank you for the comments**
> > >
> > > We thank the reviewer for the feedback and for re-evaluating the score.
> > >
> > > Regarding the comments on Table 1 and 2:
> > >
> > > (1)	The “Nowozin Trick” as named in the WAE-GAN code (on Github) by Tolstikhin et al. is the idea proposed in the Adversarial Variational Bayes [AVB] paper and it goes as follows. To perform adversarial discrimination between the encoded distribution Pz and the prior distribution Qz, one needs an optimal discriminator D_JS(Pz,Qz), (e.g. based on the Jensen-Shannon divergence), which is give by Dopt(x)=log(qz(x))- log(pz(x)) where for analytic qz we exactly know qz(x) (i.e. for a Gaussian qz). Therefore, it is valid to add log(qz(x)) explicitly to the discriminator and let it learn only the remaining term, log(pz(x)). In short, although this trick is useful for training, it requires analytic knowledge of the latent prior. Since a primary point of our paper is that SWAE does not require any such knowledge, we believe it was fair to disable it, as to facilitate.
> > >
> > > (2)	As for the FID scores, we used the WAE-GAN code, and ran it without the `Nowozin Trick’ on the CelebA dataset and are reporting the results we got from the code. The discrepancy could be due to several mismatches between the runs including, the si called ‘Nowozin Trick’, number of training iterations, and the preprocessing of the data.
> > >
> > > (3)	Regarding the log likelihood scores, we measure the likelihood of the encoded samples \phi(x), x ~ pX, to be generated from qZ, which is calculated as follow E_pX[ log(qZ(\phi(x)) ]. We emphasize that this measure is only valid for prior distributions, qZ, with analytic form.
> > >
> > >
> > > We also just became aware of another submission to ICLR2019, “Cramer-Wold AutoEncoder” (CWAE) https://openreview.net/forum?id=rkgwuiA9F7 , in which the authors cite our work and provide an extensive comparison of our method (SWAE) to WAE and their proposed framework (CWAE). This comparison was possible as we released our code in April 2018. The authors of CWAE kindly provide further quantitative results (See Figure 4 of their paper) for SWAE and WAE on CelebA (and for CIFAR10 in the supplementary material) comparing the FID score, and Mardia’s skewness and kurtosis of models WAE, SWAE and CWAE, on the CelebA test set. CWAE as the authors put it “can be seen as a borderline model between SWAE and WAE-MMD”, “which has a closed-form for the distance of a sample from standard multivariate normal distribution.”
> > >
> > > We hope that the extensive analysis provided by the authors of CWAE, in addition to our extended experimental section and numerical analysis, to further convince the reviewer on the merit of our proposed method.
> > >
> > > [AVB] Mescheder, L., Nowozin, S. and Geiger, A., 2017. Adversarial variational bayes: Unifying variational autoencoders and generative adversarial networks. arXiv preprint arXiv:1701.04722.

---

### Author Response · Authors · 2018-12-12
**Request for Re-evaluation**

Dear Reviewers,

We are certainly grateful for your time and careful evaluation of our work.  We did our best to respond to the issues you raised, including extending our experimental results along with theoretical clarifications.

We would greatly appreciate if you could take a second look at our paper and reevaluate based on the changes, our responses, and the additional information we provided regarding the “Cramer-Wold AutoEncoder” (CWAE) paper https://openreview.net/forum?id=rkgwuiA9F7 .

Thank you in advance for your time.

---

### Meta-Review · Area_Chair1 · 2018-12-14
**Empirical advantage is not very clear but the idea and the theoretical analysis are intersting.**

**Confidence:** 2
**Recommendation:** Accept (Poster)

**Metareview:**

The paper proposed to add the sliced-Wasserstein distance between the distribution of the encoded training samples and a samplable prior distribution to the auto encoder (AE) loss, resulting in a model named sliced-Wasserstein AE. The difference compared to the Wasserstein AE (WAE) lies in using the usage of the sliced-Wasserstein distance instead of GAN or MMD-based penalties.
The idea of the paper is interesting, and a theoretical and an empirical analysis supporting the approach are presented. As reviewer 1 noticed, „the advantage of using sliced Wasserstein distance is twofold: 1)parametric-free (compared to GANs); 2) almost hyperparameter-free (compared to the MMD with RBF kernels), except setting the number of random projection bases.“ However, the empirical evaluation in the paper and concurrent ICLR submission on Cramer-World-AEs the authors refer to shows no clear practical advantage over the WAE, which leads to better results at least regarding the FID score. On the other hand, the Cramer-World-AE is based on the ideas presented in this paper (which was previously available on arxive) proving that the paper presents interesting ideas which are of value to the communty. Therefore, the paper is a bit boarderline, but I recommand to accept it.